# Influencers on Thyroid Cancer Onset: Molecular Genetic Basis

**DOI:** 10.3390/genes10110913

**Published:** 2019-11-08

**Authors:** Berta Luzón-Toro, Raquel María Fernández, Leticia Villalba-Benito, Ana Torroglosa, Guillermo Antiñolo, Salud Borrego

**Affiliations:** 1Department of Maternofetal Medicine, Genetics and Reproduction, Institute of Biomedicine of Seville (IBIS), University Hospital Virgen del Rocío/CSIC/University of Seville, 41013 Seville, Spain; berta.luzon@ciberer.es (B.L.-T.); raquelm.fernandez@juntadeandalucia.es (R.M.F.); leticia.villalba.benito@hotmail.es (L.V.-B.); ana.torroglosa@juntadeandalucia.es (A.T.); guillermo.antinolo.sspa@juntadeandalucia.es (G.A.); 2Centre for Biomedical Network Research on Rare Diseases (CIBERER), 41013 Seville, Spain

**Keywords:** thyroid cancer, genetics, epigenetics, mutation

## Abstract

Thyroid cancer, a cancerous tumor or growth located within the thyroid gland, is the most common endocrine cancer. It is one of the few cancers whereby incidence rates have increased in recent years. It occurs in all age groups, from children through to seniors. Most studies are focused on dissecting its genetic basis, since our current knowledge of the genetic background of the different forms of thyroid cancer is far from complete, which poses a challenge for diagnosis and prognosis of the disease. In this review, we describe prevailing advances and update our understanding of the molecular genetics of thyroid cancer, focusing on the main genes related with the pathology, including the different noncoding RNAs associated with the disease.

## 1. Introduction

Thyroid cancer is the most common endocrine cancer, with an increasing overall incidence in recent decades of about two fold in the last 25 years and accounting for 2% of all cancers [1]. Thyroid cancer is the sixth most common cancer in women, who are three times more likely to have thyroid cancer than men. About 2% of cases occur in children and teens. Overall, the 5-year survival rate of thyroid cancer is 98%. However, survival rates depend on many factors, such as the specific type of thyroid cancer and stage of disease [2]

Several types and histological subtypes can be found depending on the cells from which the tumor derives, and each one presents different characteristics and prognoses. Non-medullary thyroid cancer (NMTC) originates from follicular cells and is responsible for ~95% of all the cases. At the same time, NMTC is divided into four groups (also known as well differentiated thyroid cancer): (1) papillary thyroid cancer (PTC), which represents more than 85% of cases, (2) follicular thyroid cancer (FTC; 10% of total cases), (3) poorly differentiated thyroid cancer (PDTC, which accounts for only 1–15% of all thyroid cancers), constitutes a more aggressive follicular-derived thyroid cancer than differentiated thyroid cancer (PTC and FTC), and (4) anaplastic cancer (ATC) (<1%) [3].

Of the total number of cases, 10% of NMTCs occur during the first two decades of life [4]. Around 5% present as familial forms and the remaining 95% are sporadic. Analysis (genome-wide association study or genome-wide linkage analysis) has led to the identification of some genes associated with non-syndromic familial forms, although additional populations are needed for further validations [5,6,7,8,9]. In either case, the studies carried out to date are quite inconclusive, and thus new approaches should be carried out for the identification of new loci associated with FNMTC.

In addition, only ~5% of cases of thyroid cancer are derived from parafollicular cells and this type is named medullary thyroid cancer (MTC) [4,10]. About 75% of all MTCs are believed to be sporadic (sMTC), and the remaining 25% correspond to inherited cancer syndromes known as multiple endocrine neoplasia type 2 (MEN2). MEN2 includes three clinically differentiable types: MEN2A (OMIM #171400), MEN2B (OMIM #162300), and FMTC (OMIM #155240; which may be a variant of MEN 2A) [11,12]. FNMTC is a rare form of thyroid cancer (only 3–9% of all cases). Moreover, just 5% of the family forms are included in specific syndromes: Cowden (OMIM #158350), Gardner (OMIM #175100), Werner (OMIM #277700), Li-Fraumeni (OMIM #151623), McCune-Albright (OMIM #174800), Carney complex (OMIM #160980), or DICER 1 (OMIM #138800) syndromes [5,6]). This review summarizes the current knowledge about molecular bases of thyroid cancer: from genes to ncRNAs, to provide the reader an overview of the main genetic and epigenetic factors that can influence the develop of thyroid cancer.

## 2. Molecular Genetics of Thyroid Cancer

Alterations on some genes and pathways have been described to be associated with the different forms of thyroid cancer. The most relevant are summarized in the current review (Figure 1). Thyroid tumorigenesis and progression are closely linked to somatic point mutations on *BRAF* (B-rapidly accelerated fibrosarcoma), *RAS*, *RET*, and *NTRK1/3* genes that promote the activation of mitogen-activated protein kinase (MAPK) and phosphoinositide 3-kinase (PI3K) signaling pathways [13,14]. Interestingly, there is a strong relationship between the oncogenic genotype and histopathologic phenotype, mutations in *BRAF*, *PAX8*, and *PPARG* genes being common in FTC and *RAS* mutations either in FTC and PTC (follicular variant, fvPTC) [15,16,17,18,19]. The global analysis of genomic variants, gene and miRNA expression, aberrant methylation, and proteomic profiles unraveled that mutations in *BRAF* and *RAS*, together with *RET-PTC* fusions, are the most common genetic alterations in both FTC and PTC.

It is worth mentioning The Cancer Genome Atlas (TCGA) project outcomes, where a comprehensive and multiplatform approach to analyze multidimensional molecular data of 496 PTCs (excluding both poorly and undifferentiated carcinomas to compile all tumor-initiating alterations) was performed. Such a cohort led to classification into molecular subtypes (including tumor cluster, histology, genotype, signaling, and risk assessment) to associate them with clinically relevant parameters and improve the management of patients. They mainly distinguish two types of PTC tumors: PTCs with RAS mutations (RL-PTCs) and BRAFV600E-driven tumors (BVL-PTCs). Both are basically different in their genomic, epigenomic, and proteomic profiles, which is in concordance with their histological differences [20].

## 3. The *RET* Proto-Oncogene Role

In 1993 *RET* proto-oncogene was demonstrated to be the causative event for MTC [21]. Since then, several mutations have been described in MEN2 series. Thus, the *RET* genetic screening is a crucial diagnostic and prognostic procedure for MTC [22].

### 3.1. Somatic *RET* Mutations

#### 3.1.1. *RET* Gene Rearrangements

The Rearranged During Transfection (*RET*) gene encodes a single transmembrane receptor, which is a tyrosine kinase implicated in the cellular signaling of members of the glial derived neurotrophic factors family [23].

In PTC, *RET* rearrangements (*RET-PTC*) seem to be an early event in carcinogenesis, with 10–20% of *RET* fusions being detected in PTC patients. *RET-PTC* fusions used to be associated with sporadic and radiation-induced PTC [24,25]. In addition, *coiled-coil domain containing 6-RET* rearrangements are also commonly found in this type of tumor [26,27]. The presence or higher expression of a RET fusion protein has been linked to regional invasion and lymph node metastasis [28].

Thus, *RET* rearrangements are determinant on the evaluation of thyroid nodules with undefined cytology. The main RET/PTC oncoproteins are summarized in Table 1.

#### 3.1.2. Other Somatic RET Mutations

The genetic bases for nearly all thyroid cancers have been solved through DNA sequencing studies. Most thyroid tumors harbor mutations that eventually activate MAPK and PI3K-AKT signaling pathways, which regulate cellular proliferation [3,40,41]. Furthermore, somatic *RET* mutations appear in 40–50% of sporadic MTCs and have been linked with a worse prognosis for patients. In fact, the presence of a somatic *RET* mutation is associated with lymph node metastases at diagnosis, which is a bad prognostic factor for complete cure for such patients [22]. Somatic *RET* mutations have been described in sporadic cases of MTC [42] as well as loss of heterozygosity at different loci (deletions of tumor suppressor genes) in MTC [21]. In addition, a mutation in codon 664 of the *RET* gene, which affects the protein kinase domain of the gene product by changing a threonine for a methionine, was found in unrelated MEN2B patients as well as in six sporadic tumors [43].

The application of new approaches has improved the knowledge of somatic RET mutations such as *RET*^M918T^ mutation. A genome-wide DNA methylation profiling was performed in a large MTC series. Integration of methylation data together with mRNA and miRNA expression data revealed JAK/Stat signaling effector STAT3 as a potential therapeutic target for these tumors [44].

### 3.2. Germline RET Mutations

Gain-of-function *RET* mutations give rise to MEN2 [23,45]. In MEN2, mutations in the RET intracellular domains lead to activation of RET monomers while in the extracellular domain carry to dimerization [23,46]. Thus, MEN2 RET receptors constitutively activate signaling pathways linked to wildtype RET activity [47,48]. In fact, MEN2 phenotype is more severe when specific *RET* mutations lead to increased kinase activity [49,50]. Based on these observations, the clinical management guidelines for MEN2 patients recommend early genetic screening in at-risk individuals, in order to determine if they carry any *RET* mutation, which is associated with the poorest prognosis [50]. There are three American Thyroid Association categories: I) moderate-risk mutations (exon 10, exon 11 mutations other than codon 634, and exons 13 through 15), which lead to relatively late MTC onset; II) high-risk mutations (exon 11, codon 634, the classical MEN2A mutation; and exon 15, codon 883), which present intermediate risk; and III) highest-risk mutations, with the typical MEN2B mutation (exon 16, codon 918), which confer the highest risk of early MTC development and growth [51]. All updated information regarding the effect of *RET* mutations on sporadic and familial cases of thyroid cancer is compiled in different reviews [26,52,53].

More than 95% of MEN2 cases have germline mutations in the *RET* proto-oncogene, leading to a constitutive activation of the metabolic pathways of *RET* signaling [54]. Of those patients, 98% have mutations grouped in a hotspot composed of five cysteine codons present in the extracellular domain of the protein (C609, C611, C618, C620, C630, C634) [49,55,56]. Eighty-seven percent of MEN2A patients carry mutations affecting codon 634, C634R being the most prevalent one that has been detected [49], unlike in the Spanish population, who mostly carry the C634Y mutation [57,58,59,60]. In addition, although over 70 *RET* mutations have been described in MEN2A patients (all in the extracellular region, close to the transmembrane domain), only two *RET* mutations (affecting codons 918 and 883) have been linked to MEN2B [61].

Regarding FMTC missense mutations in the tyrosine kinase domain, one changes glutamate 768 for an aspartate and another one substitutes valine 804 for a leucine. In addition, there are some relevant mutations on cysteine codons and the transmembrane domain, some of them shared with MEN2A syndrome, which are updated and summarized in Figure 2. Both are gain-of-function mutations and result in aberrant signaling mediated by RET [42,62,63].

### 3.3. Susceptibility Factors

A germline *RET* S836S variant (c.2508C>T, rs1800862) has been correlated with somatic *RET*^M918T^ mutation in exon 16, which leads to a more aggressive sMTC. Furthermore, this variant was found to have a significantly higher frequency in sMTC compared to controls, which led to it being considered as the first genetic susceptibility factor for the disease [64,66,67,68]. Additionally, IVS1-126G>T (c.74-126G>T, rs2565206) and G691S (c.2071G>A, rs1799939, p.Gly691Ser)/S904S (c.2712C>G, rs1800863) have also been described as associated with sMTC in several populations although this finding was not replicated in a larger study [69]. The *RET* c*587T>C variant was restricted to the Italian population [70,71,72,73].

It has been suggested that variants in genes encoding for *RET* coreceptors might play a role in the pathogenesis of sMTC. A study of the German population showed that GFRA1-193C>G and c.537T>C were overrepresented in sMTC versus controls, which suggested that both variants could be in linkage disequilibrium with other locus responsible for the disease. However, such overrepresentation was not found in a similar study of the Spanish population, supporting a founder effect of both variants in Germany [74].

## 4. *BRAF*

The most frequent mutations (60%) in PTC patients are found on the *BRAF* gene. Moreover, the *BRAF^V600E^* mutation (p.Val600Glu; commonly known as V600E), which is found in 95% of cases, is used as a risk biomarker in PTC [20]. Although somatic mutations in this gene are absent on benign thyroid nodules, they appear on a third of anaplastic thyroid cancers [75,76]. As we mentioned above, the majority of *BRAF* mutations implicate the activation of the kinase domain of this protein, which finally dysregulates the MAPK signaling pathway [77,78,79], thus being relevant in cellular processes implicated in cancer development such as tumorigenesis, proliferation, and progression [80,81].

## 5. *RAS*

The *RAS* genes (*H-RAS*, *N-RAS*, *K-RAS*) present somatic point mutations in follicular adenoma, FTC (40–53%), PTC (0–20%), fvPTC (17–25%), and poorly differentiated and anaplastic thyroid cancer (20–60%) [19,75,82,83,84,85]. A codon 61 mutation of *NRAS* (N2) was identified four times more frequently in follicular tumors than in papillary cancers, and it is the second most common point mutation with an incidence of 8.5% [20]. Thus, *RAS* mutations are associated with follicular tumors that compile from a preinvasive lesion to a true malignancy, either FTC, PTC, fvPTC, or poorly differentiated thyroid cancer [84,86]. *RAS* mutations together with *TERT* promoter mutations (C228T and C250T) have been associated with more aggressive and recurrent thyroid tumor and patient mortality, especially in PTC patients. Although further studies are needed, these outcomes indicate that *TERT* is a new oncogene in thyroid cancer and promoter mutations may be promising in clinical management of thyroid cancer, especially in combination with BRAF V600E or RAS mutations [87].

## 6. PAX8-PPARG

The PAX8-PPARG fusion protein is the product of a t(2;3)(q13;p25) chromosomal translocation, which used to be linked to FTC, and presents oncogenic capacity in transgenic mice [88]. *PAX8-PPARG*-positive tumors present a more prominent vascular and capsular invasion than *RAS*-positive tumors [16].

## 7. Other Genes

In ATC the accumulation of several oncogenic alterations is equivalent to an increased level of dedifferentiation and aggressiveness [89]. The role that P53 plays in thyroid carcinogenesis is well known, but the role of the remaining *P53* family members in thyroid cancer needs further studies. Increasing evidence indicates that such family members favor the development of multiple thyroid cancer variants, and in addition they are being used as therapeutical targets [90].

Furthermore, there is a lack of knowledge of pathways specifically associated with each *RET* mutation and with non-*RET*-mutated sporadic MTC. A transcriptional profile assay, together with pathway enrichment analysis and gene ontology biological processes, revealed that *PROM1* could be necessary for the survival of tumoral cells in this tumor [91].

In addition, a novel constitutional frameshift c.948delT mutation (p.G318Afs*22) in *ESR2* was found to segregate with MTC (without any identifiable *RET* mutation) [92]. However, this mutation was not found in other series of patients, such as FMTC cases [93].

Rearrangements involving the anaplastic lymphoma kinase (*ALK*) gene with the striatin (*STRN*) gene have been described, which constitutively activate ALK kinase, inducing tumor formation in nude mice. Such fusion may represent a therapeutic target for patients with highly aggressive types of thyroid cancer [94]. Moreover, the *ETV6-NTRK3* rearrangement, exclusively found in fvPTC, together with *STRN-ALK* are recurrent and absent in benign lesions, which could be useful for the diagnosis of thyroid nodules [95].

There is a thyroid specific transcription factor, *FOXE1*, which plays an essential role in thyroid development. Among other functions, it recognizes sites in thyroglobulin and thyroperoxidase and it also aids in maintaining cellular differentiation in the adult thyroid. Some genes increase its expression (*Adamts9*, *Cdh1*, *Duox2*, and *S100a4*) in the absence of FOXE1, while *Casp4*, *Creld2*, *Dusp5*, *Etv5*, *Hsp5a*, *Nr4a2*, and *Tm4sf1* decrease [96]. *FOXE1* has been associated with hypothyroidism and PTC [97]. A single nucleotide polymorphism in FOXE1 rs965513 which has been related with differentiated thyroid cancers in Caucasian population [98]

A compilation of the main genes related with the disease, indicating their location, their main type of alterations (somatic or germline mutations) and their incidence on the disease has been included in Table 2.

In summary, the molecular pathogenesis of differentiated thyroid cancer, with specific signaling pathways and activating point mutations, has been elucidated. When some of the specific genes mutate, the tumor become into more aggressive and advanced thyroid cancer. Some of those events, include: *RAS* mutations (25% in poorly differentiated thyroid cancers); activating mutations of *PIK3CA*; point mutation p.V600E in *BRAF*; mutations in *CTNNB1* (β-catenin) which is involved in Wnt signaling, and in *TP53* (a tumor suppressor), which may participate in de-differentiation of these tumors [116,117]. In addition, copy-number gains (found in proto-oncogenes) or oncogenic gene amplifications are mostly found in poorly differentiated and anaplastic than in differentiated thyroid carcinomas, suggesting that these genetic alterations play an important role in the progression and aggressiveness of thyroid cancer. This is the case, for example, of the genes encoding receptor tyrosine kinases (RTKs) and *P13K-AKT* pathway members, where an increased protein expression will activate these signaling pathways [118]. Regarding *ALK* gene, it was found in 9% of poorly differentiated thyroid cancers, 4% of ATC and 1% of PTC [119]. While *MAPK* and *P13K-AKT* pathways are firstly involved in differentiated thyroid carcinoma, when genetic alterations accumulate, both pathways get activated and an evolution of the tumor occurs into poorly differentiated and ATC [14].

## 8. ncRNAs

The global analysis of genomic variants, gene and microRNA (miRNA) expression, aberrant methylation, and proteomic profiles revealed that mutations in *BRAF* and *RAS*, together with *RET-PTC* fusions, are the most common genetic alterations in both FTC and PTC. However, epigenetic modulators, such as non-coding RNAs (ncRNAs), have recently emerged as helpful factors for both the diagnosis and treatment of these thyroid carcinomas. Specifically, ncRNAs such as miRNAs, circular RNAs (circRNAs), and long non-coding RNAS (lncRNAs) have been implicated in the modulation of gene expression that regulates cellular processes such as cell differentiation, proliferation, cell cycle, apoptosis, migration, and invasion [120,121,122,123]. Over the past four decades, thyroid cancer has emerged as a major health issue. Thus, the identification of novel molecular therapeutic targets for prognosis and diagnosis to advance in the overall management of this malignancy is really needed. Interestingly, ncRNAs have been identified together with epistatic gene interactions in sMTC and juvenile PTC, which help to understand the genetic architecture of complex diseases and support the relevance that these elements have to explain carcinoma development and progression [124,125].

### 8.1. miRNAs

In the last twenty years, dysregulation of miRNAs have been linked to thyroid dysfunction and oncogenicity leading to this type of cancer [122,126]. MiRNAs are an important class of small regulatory RNAs with a length of ~22 nt that regulate post-transcription gene silencing [127,128]. They regulate gene expression by partial complementarity pairing with mRNAs, promoting their degradation or blocking translation. This gene regulation leads them to participate in many biological processes [129,130]. When thyroid malignant transformation occurs by MAPK oncogenes, a significant reduction of “tumor-suppressor” miRNAs and activation of oncogenic miRNAs take place [131]. The most relevant thyroid differentiation genes and transcription factors as predicted targets of microRNAs have been summarized in Table 3. The role of specific miRNAs in thyroid carcinogenesis is under intensive investigation, as new examples of miRNAs in cancer pathways are reported almost daily. As an example of such complexity, TCGA analysis unravel the presence of six miRNA clusters in PTC. Cluster 1 (*RAS*-mutated tumors and follicular variants) is enriched with the miRNAs miR-181 and miR-182, while *BRAF* tumors are contained in the rest of clusters (Clusters 2–6). Cluster 5 (with high levels of miR-146b and miR-375, and low levels of miR-204) and Cluster 6 (enriched with miR-21 and low levels of miR-204), are associated with the less-differentiated tumors and a higher risk of recurrence [132]. In summary, the upregulated miRNAs target factors that suppress pathway activation (of the three main signaling pathways activated in thyroid cancer—MAPK, PI3K, and TGFβ), while the downregulated miRNAs target factors activating such pathways. In this scenario, those three pathways remain activated in thyroid cancer where miRNAs are functioning as crucial modulators of pathway activity. Then, they can be considered as potential therapeutic targets.

### 8.2. lncRNAs

More recently, many lncRNAs (>200 nt or even several kb in length) have been described to be upregulated or downregulated in thyroid cancer tissues or cancer cell lines. Predicting miRNA/target duplex [138] is used to detect miRNA/lncRNA interaction [139].

They are expressed in a tissue-specific manner and involved in tumorigenesis acting through distinct molecular mechanisms on epigenetic modifications, transcriptional, and post-transcriptional processing. In this way, they are implicated in cell cycle, cell differentiation, proliferation, apoptosis, migration, and invasion [120,140]. However, the role and molecular mechanism that they play in thyroid cancer, compared to other malignancies, remains largely unclear. Their involvement in cancer pathogenesis and availability make lncRNAs ideal biomarkers for cancer prognosis and diagnosis. Just as an example, we will discuss the importance of three relevant lncRNAs in thyroid cancer such as *H19*, *MALAT1*, *PTCSC3*, among many others whose descriptions and their possible mechanism in thyroid cancer, known to date, are summarized in Table 4.

#### 8.2.1. H19

*H19* is a typical lncRNA that is associated with various cancer types, with both tumor promoter and suppressive functions [141]. Its expression is higher in thyroid tumor tissues, where enhance proliferation and migration, comparing with normal thyroid cells [142].

#### 8.2.2. Metastasis-Associated Lung Adenocarcinoma Transcript 1 (MALAT1)

*MALAT1* is a prooncogenic lncRNA in several cancer types [143]. Regarding thyroid cancer, MALAT1 is highly expressed in PTC than in FTC and ATC [144]. In addition, *MALAT1* expression is higher in MTC than in normal thyroid [145].

#### 8.2.3. Papillary Thyroid Carcinoma Susceptibility Candidate 3 (PTCSC3)

*PTCSC3* is a lncRNA whose expression is strictly thyroid-specific and is downregulated either in thyroid tumor tissues and in thyroid cell lines. When *PTCSC3* acts as a tumor suppressor is a competing endogenous RNA for miR-574–5p [146].

**Table 4 genes-10-00913-t004:** LncRNAs associated with thyroid cancer (TC).

lncRNA	Role in Thyroid Cancer	References
RP5-1024C24.1	Associates negatively with late clinical stages	[147]
CASC2	Expression correlates with multifocality and TNM	[148]
PANDAR	Inhibits proliferation, cell cycle and promote apoptosis	[149]
ENSG00000235070.3	Correlates directly to BRAF (V600E)	[150]
ENSG00000255020.1		
GAS8-AS1	Suppresses cell proliferation in thyroid cancer	[151]
	Low Expression negatively correlates with LNM	[152]
NONHSAG051968	Correlates negatively with tumor size	[153]
NONHSAG018271	Suppresses tumor cell growth	
NONHSAG007951		
LINC00271	Involves in extrathyroidal invasion, LNM, advanced tumor stage and recurrence in TC	[154]
LINC00663	Its role in cancer formation needs further investigation	[155]
NONHSAT037832	Plays role in LNM and determines tumor size	[156]
MEG3	Inhibits invasion and associates with LNM	[157]
PTCSC2	Predisposes genetically to thyroid cancer	[158]
PTCSC3	Suppresses cell growth and invasion	[159,160]
PTCSC1	A candidate susceptibility gene for PTC	[161]
NAMA	Targets MAPK signaling pathway	[162,163]
NONHSAT076754	Correlates to LNM in PTC	[164]
n340790	Accelerates TC cell growth, motility and inhibit apoptosis	[165]
HOTAIR	Triggers cell growth and invasion	[166]
	Associates with poor survival of TC patients	[167]
NEAT1	Promotes tumor progression and tumor size	[168]
ENSG00000273132.1	Overexpression correlates directly to BRAF (V600E)	[150]
ENSG00000230498.1	Overexpression correlates directly to BRAF (V600E)	
CTD-3193013	A node of co-regulation with other lncRNAs and tumor size	[147]
AC007255.8	Correlates to clinical stage (patient age)	
HOXD-AS1	Correlates to clinical stage	
RP11-40216.1	Expression proportional to LNM	
HIT000218960	Correlates to TNM stage, LNM, and multifocality	[169]
MALAT1	Regulates proliferation, migration and EMT via TGF-β	[144]
NR_036575.1	Promotes proliferation and migration of thyroid cancer	[170]
ANRIL	Proliferation, invasion and metastasis via TGF-β/Smad	[171]
XLOC_051122	Oncogenic with metastatic potentials	[172]
XLOC_006074	Oncogenic with metastatic potentials and prognostic role	
LOC100507661	Enhances proliferation, migration and invasion	[173]
H19	Increases proliferation, migration and invasion	[142]
FAL1	Associates positively with risk of multifocality	[174]
ENST00000537266	Promotes proliferation and inhibit apoptosis	[175]
ENST00000426615	Regulates proliferation, migration, apoptosis and cell cycle	
PVT1	Increases thyroid cancer cell proliferation	[176]
BANCR	Enhances thyroid cancer cell proliferation and inhibits apoptosis	[163,177]
FOXD2-AS1	Contribute to proliferation, migration and invasion of cancer cells, and its deregulation is related to carcinogenesis, overall survival, disease free survival, prognosis and tumor progression	[178]
AFAP1-AS1	The dysregulated expression of AFAP1-AS1 is related to carcinogenesis, overall survival, disease-free survival, progression-free survival and tumor progression containing lymph node metastasis, distant metastasis, histological grade, tumor size and tumor stage	[179]
ENST00000489676	It influences PTC cell proliferation and invasion through regulating miR-922	[180]
LUCAT1	The overexpression of LUCAT1 is related to PTC development, through acting in cell-cycle regulation, proliferation, epigenetic modifications through LUCAT1/ CDK1/ EZH2/ P57/ P21/ HDAC1/ DNMT1/ P53/ BAX axis and apoptosis, via extrinsic pathway activating caspases	[181]

Notes: LNM, lymph node metastasis; N/A, data not available; TNM, tumor node metastasis.

### 8.3. circRNAs

circRNAs were identified in human papillary thyroid cancer [182] and lately associated with different cancers [183,184,185]. There are four types: exonic circRNAs, circular RNAs from introns, exon–intron circRNAs, and intergenic circRNAs [186]. They are expressed in thousands of human genes and they have been suggested to modify gene expression through sponging miRNAs and interact with RNA-binding proteins, and they can also be positive regulators of their parental genes [187,188,189]. All circRNAs linked to thyroid cancer known to date have been compiled in Table 5.

This is a very recent field of study in thyroid cancer and then, most of the circRNAs included in Table 5 are exclusively associated with this type of tumor, at least until today. Further studies are required to decipher if they are involved in the develop of some other tumors. However, four circRNAs have already been found in other cancers: circZFR, circ-ITCH, hsa_circ_0004458 and hsa_circRNA_100395.

## 9. Conclusions and Future Perspectives

Thyroid cancer is a disease induced by progressive genetic and epigenetic molecular abnormalities. Most studies describe several classical aberrations involved in the tumorigenesis and progression of this tumor, such as point mutations, gene copy number variations, and aberrant gene methylation. Additionally, increasing evidence points to ncRNAs as important players in the regulation of a variety of biological functions of thyroid cancer cells (RNA stability, epigenetic processes, chromatin accessibility, translation, gene expression in mitochondria, anterograde and retrograde signaling, and intracellular and intercellular signaling) [173,175,204]. However, only a few structures of lncRNA are known. Furthermore, ncRNAs may be used for diagnosis, prognosis, and as potential therapeutic agents of thyroid cancer.

The genetic landscape of thyroid cancer is growing rapidly. Therapies that are targeted at genetic mutations related to this tumor are being used in patient care and clinical trials. In comparison with traditional disease-specific trials, molecular allocation studies (“basket studies”) are testing targeted therapies in patients whose tumors carry particular mutations, in spite of the tumor type or origin [205]. These studies are more informative on anticancer therapies because they include a greater number of patients. Regarding thyroid cancers due to low prevalence mutations (i.e.: rearrangements of RET, NTRK1, NTRK3, or ALK), they could be included in trials with selective kinase inhibitors with probed efficacy in other types of cancer. Thus, targeted therapy in thyroid cancer has been mainly focused on tyrosine kinase inhibitors. Furthermore, cancer adaptation to hypoxia and angiogenesis through the transcription factor hypoxia-inducible factor 1-alpha has been found [4]. This factor is only present in cancer tissues and therefore could be a precision medicine paradigm for further research focused on drivers uniquely present in cancer tissues.

Finally, the correlation of molecular profile (obtained by new sequencing technologies) with clinicopathological characteristics will lead to the development of novel precision therapies, reducing morbidity and mortality in thyroid cancer patients [206,207,208]. In this sense, differential response to a mitogen-activated protein kinase inhibitor was found to be related to the thyroid cancer genotype [209]. Molecular testing of mutation hotspots, rearrangements, and gene expression is now an effective diagnostic tool to better select patients for potential thyroid surgery [210,211].

The TCGA project mentioned in this review [20] shifts the focus regarding thyroid cancer classification to a strong multidimensional genomic and integrative approach, which has relevant consequences on basic pathobiology, tumor classification, and therapies. In this manner, thyroid cancer therapy would enter the realm of precision medicine, which will favor a more personalized approach to this pathology, achieving, ultimately, greater hope and quality of life for patients.

In summary, the diagnosis and treatment of thyroid cancer is nowadays influenced by the molecular characterization of thyroid cancer types. The improvement of such techniques would help to reduce unnecessary treatments in indolent thyroid cancers, as well as improve the outcomes in patients with clinically aggressive cancers.

## Figures and Tables

**Figure 1 genes-10-00913-f001:**
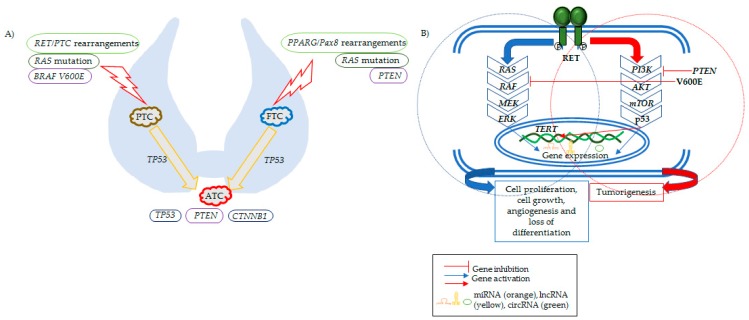
Progression of thyroid cancer and the main genetic alterations involved in such process. (**A**) Picture depicting thyroid cancer hystotypes and their causative genetic events. PTCs present BRAF (V600E substitution), RAS mutations, and/or RET/PTC rearrangements. FTCs display PPARc/Pax8 rearrangements, RAS mutations, and PTEN inactivating mutations or deletions. ATCs are characterized by PTEN and CTNNB1 mutations and p53 inactivation. (**B**) Schema shows the key molecular signaling pathways involved in thyroid cancer. On the left (inside the circle with dashed blue line): MAPK pathway, which is activated in most thyroid cancers after a mutational event. Once thyroid cancer development is initiated, gene expression is altered, evoking cell proliferation, cell growth, angiogenesis, and loss of differentiation. On the right (inside the circle with dashed red line): pathways altered in advanced thyroid cancers, which promote tumor progression. This includes the PI3K–mTOR pathway, the p53 tumor suppressor, and alterations in the promoter for TERT. miRNAs, lncRNAs and circRNAs are also represented inside the nucleus as genetic players in thyroid cancer development.

**Figure 2 genes-10-00913-f002:**
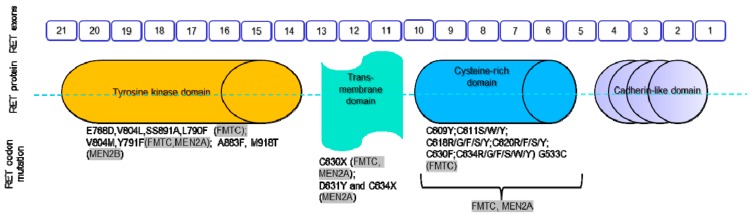
Schematic structure of RET, and mutations identified in MEN2A, MEN 2B and familial medullary thyroid carcinoma (FMTC). Modified from [64,65].

**Table 1 genes-10-00913-t001:** RET/PTC oncoproteins related with thyroid cancer.

RET/PTC Rearrangement	Fusion Partner	Chromosomal Alteration	Reference
RET/PTC1	Coiled-coil domain-containing protein 6 (CCDC6)	inv10(q11.2;q21)	[29]
RET/PTC2	Protein kinase, cAMP-dependent, regulatory, type I, alpha (PRKAR1A)	t(10;17)(q11.2;q23)	[30]
RET/PTC3	Nuclear coactivator 4 (RET-fused gene, androgen receptor-associated protein 70) (NCoa4 (RFG, ARA70)	inv10(q11.2;q10)	[29]
RET/PTC4	Nuclear coactivator 4 (RET-fused gene, androgen receptor-associated protein 70) (NCoa4 (RFG, ARA70)	inv10(q11.2;q10)	[31]
RET/PTC5	RET-fused gene 5 (Golgin A5) (RFG5 (GOLGA5))	t(10;14)(q11.2;q32)	[32]
RET/PTC6	Tripartite motif-containing 24 (TRIM24)	t(7;10)(q32-q34;q11.2)	[33]
RET/PTC7	RET-fused gene 7 (tripartite motif-containing 33 (RFG7(TRIM33))	t(1;10)(p13;q11.2)	[33]
RET/PTC8	Kinectin 1 (KTN1)	t(10;14)(q11.2;q22.1)	[34]
RET/PTC9	RET-fused gene 9 (RFG9)	t(10;18) (q11.2;q21-22)	[35]
RET/ELKS	Glutamate, leucine, lysine, serine-rich sequence (RAB6-interacting protein 2) (ELKS (RAB6IP2))	t(10;12)(q11.2;p13.3)	[36]
RET/PCM1	Pericentriolar material 1 (PCM1)	t(8;10)(q21-22;q11.2)	[37]
RET/RFP	RET finger protein (tripartite motif-containing 27) (RFP (TRIM27))	t(6;10)(p21;q11.2)	[38]
RET/HOOK3	Hook homolog 3 (HOOK3)	t(8;10)(p11.21;q11.2)	[39]

Most relevant genes in thyroid cancer: a compilation of the main genes related with the disease, indicating their location, their main type of alterations (somatic or germline mutations), and their incidence on the disease. A footnote with the main databases available to access all RET variants “with clinical relevance to date”, indicating their number registered on each database until the last access, is included.

**Table 2 genes-10-00913-t002:** Most relevant genes in thyroid cancer.

Gene	Chromosome (Chr)	Type of Alteration	Origin of Mutation	Disease	Reference
*RET* (*)	Chr 10	RET/PTC rearrangements	Somatic	PTC	[26,27,28]
		Point mutations	Somatic	sporadic MTC	[43,99]
			Germline	MEN2A [OMIM# 171400]), MEN2B [OMIM#162300]) and FMTC [OMIM# 155240])	[23,45]
*BRAF*	Chr 7	V600E mutation (p.Val600Glu)	Somatic	PTC	[75,76]
		Point mutations	Somatic	ATC	
*RAS (N-, K-, H-)*	*NRAS:* Chr 1; *KRAS*: Chr 12; *HRAS*: Chr 11	Point mutations	Somatic	follicular adenoma, FTC, PTC, fvPTC, poorly differentiated and anaplastic thyroid cancer	[19,75,82,83,84,85]
*PTEN*	Chr 10	Insertions, deletions, splice site mutations and large deletions	Germline	Cowden syndrome 1 (CWS1, [OMIM# 158350]	[100]
*PIK3CA*	Chr 3	Point mutations	Somatic	PTC	[101]
			Germline	Cowden syndrome 5 (CWS5, [OMIM# 615108])	[17]
*AKT1*	Chr 14	Point mutations	Germline	Cowden syndrome 6 (CWS6, [OMIM# 615109]	[102]
*TERT Promoter*	Chr 5	Mutations with a co-occurrence with either *BRAF* or *RAS* mutations	Somatic	ATC and advanced stages of FNMTC	[103,104]
*JAK3*	Chr 19	Point mutations	Somatic	FTC	[105]
*TP53*	Chr 17	Point mutations	Somatic	ATC, PDTC	[106,107]
			Germline	Li-Fraumeni syndrome (LFS, [OMIM# 151623])	[108]
*CHEK2*	Chr 22	Deletions and point mutations	Germline	Li-Fraumeni syndrome 2 (LFS2, [OMIM# 609265])	[109]
*MET*	Chr 7	Point mutations	Somatic	MTC	[110]
*ALK*	Chr 2	Gene rearrangements	Somatic	PTC, PDTC and ATC	[111]
*APC*	Chr 5	Point mutations	Somatic	PTC	[112,113]
			Germline	Gardner syndrome [OMIM# 175100]	[113,114]
*CTNNB1*	Chr 3	Point mutations	Somatic	PTC	[115]

(*) Databases including the registered *RET* variants up to date (October 25th, 2019): LOVD (v.3.0 Build 21c): https://databases.lovd.nl/shared/genes/RET. From the 146 *RET* variants reported, this database does not distinguish which ones are linked to thyroid cancer. Then, in order to see the specific variants associated with thyroid cancer, it is useful to visit https://databases.lovd.nl/shared/diseases#id=0&order=symbol%2CASC&search_name=thyroid%20cancer&page_size=100&page=1. ClinVar: https://www.ncbi.nlm.nih.gov/clinvar/?term=RET%5Bgene%5D+thyroid+cancer. There have been reported 813 RET variants linked to any form of thyroid cancer. HGMD: http://www.hgmd.cf.ac.uk/ac/gene.php?gene=RET. From the 395 registered RET mutations in this database, 133 of them are associated to thyroid cancer.

**Table 3 genes-10-00913-t003:** Targets of microRNAs: different genes and transcription factors have been described as targets of miRNAs.

Target	miRNA										References
SLC5A5 (NIS)	miR-10-5p	miR-15-5p	miR-16-5p	miR-21-5p	miR-24-3p	miR-33-5p	miR-129-3p	miR-153	miR-195-5p	miR-199-3p	[131,133,134,135,136,137]
		miR-200-3p	miR-212-35p	miR-223-3p	miR-383-5p	miR-424	miR-429	miR455-3p	miR-489-3p	miR-590-5p	
TG	miR-10-5p	miR-18-5p	miR-24-3p	miR-25-3p	miR-30-5p	miR-32-5p	miR-33-5p	mR-92-3p	miR-133a-3p	miR-133b	
	miR-138-5p	miR-140-3p	miR-142-5p	miR-145-5p	miR-204-5p	miR-211-5p	miR-216-5p	miR-338-3p			
TPO	miR-28-3p	miR-99-5p	miR-100-5p	miR-132-3p	miR-212-3p	miR-212-5p	miR-370-3p	miR-543			
TSHR	miR-19-3p	miR-33-5p	miR-140-3p	miR-181-5p	miR-219-5p	miR-382-3p	miR-493-5p	miR-506-5p	miR-508-3p		
SLC26A4 (Pendrin)	miR-10-5p	miR-17-5p	miR-18-5p	miR-19-3p	miR-20-5p	miR-26-5p	miR-33-5p	miR-96-5p	miR-103-3p	miR-107	
	miR-122-5p	miR-129-5p	miR-133a-3p	miR-144-3p	miR-148-3p	miR-150-5p	miR-181-5p	miR-183-5p	miR-200-3p	miR-302	
SLC16A2 (MCT8)	miR-7-5p	mR-9-5p	miR-19-3p	miR-22-3p	miR-23-3p	miR-24-3p	miR-29-3p	miR-31-5p	miR-24-5p	miR-138-5p	
	miR-143-3p	miR-145-3p	miR-150-5p	miR-182-5p	miR-191-5p	miR-200-3p	miR-338-3p	miR-375	miR-365-3p	miR-429	
DUOX1	miR-34-5p	mir-125-5p	miR-134-5p	miR-135-5p	miR-151-3p	miR-153-3p	miR-199-3p	miR-217	miR-324-5p	miR-338-3p	
	miR-371-5p	miR-376c-3p	miR-448	miR-449-5p	miR-491-5p	miR-532-3p					
DUOX2	miR-9-5p	miR-15-5p	miR-16-5p	miR-18-5p	miR-31-5p	miR-34-5p	miR-135-5p	miR-140-5p	miR-148-3p	miR-150-5p	
	miR-152-3p	miR-192-5p	miR-195-5p	miR-203a-3p	miR-205-5p	miR-223-3p	miR-424-5p	miR-455-3p	miR-489-3p	miR-497-5p	
PAX8	let-7-5p	miR-24-3p	miR-34-5p	miR-101-3p	miR-122-5p	miR-137	miR-138-5p	miR-140-3p	miR-146-3p	miR-150-5p	
	miR-182-5p	miR-212-5p	miR-214-5p	miR-221-3p	miR-222-3p	miR-302-3p	miR-372-3p	miR-373-3p	miR-449-5p	miR-520-3p	
NKX2-1	let-7-5p	miR-15-5p	miR-16-5p	miR-23-3p	miR-24-3p	miR-30-5p	miR-129-5p	miR-133a-3p	miR-133b	miR-138-5p	
	miR-150-5p	miR-196-5p	miR-199-5p	miR-214-5p	miR216-5p	miR-223-3p	miR-338-3p	miR-365-3p	miR-375	miR-424	
FOXE1	let-7-5p	miR-1	miR-10-5p	miR-17-5p	miR-20-5p	miR-23-5p	miR-98-5p	miR-128-3p	miR-129-5p	miR-130-3p	
	miR-138-5p	miR-140-3p	miR-146-5p	miR-155-5p	miR-182-5p	miR-190-5p	miR-194-5p	miR-302	miR-372-3p	miR-373-3p	
HOXB4	MiR-10a	miR-10b	miR-23a	miR-218							
TIMP3	miR-221/222										
ZNFR3	miR-146b-5p										
FN1	miR-16										
ITGA2	miR-613										
PTEN	miR-21	miR-107	miR-146b	miRs 221/222							
AXIN2	hsa-miR-15a										
TP53INP1	hsa-miR-20b	hsa-miR-106a									
TP53INP2	hsa-miR-20a										
BCL2	hsa-miR-15a										
KAT2B	hsa-miR-20a										
PTEN	miR-15a	miR-19a	miR-19b	miR-21	miR-24	miR-107	miR-146b p	miRs 221/222	miR-486-5p		
KIT	miRs 221/222										
AKT3, ZNRF3, Smad4	miR-145										
p27Kip 1	miRs 221/222										
SOCS4	miR-25										
Sox17	miR-595										
Pdcd4	miR-21	miR-183									
ZEB1 and ZEB2	miR-200 family										
LOX	miR-29a	miR-30a									
SLC7A5, ADAM9, LRP6, CXCR4	miR-126										
EGFR, CXCL 12	miR-137										
MRTF-A	miR-206										
Rac1	miR-101										
SphK2	miR-613										
CARMA 1	miR-539										
TUSC2, Rock1	miR-584										
YAP1, SLC16a2, ERBB2	miR-375										
VEGF-A	miR-205	miR-126									
p85b	miR-126										

**Table 5 genes-10-00913-t005:** CircRNAs described on thyroid cancer.

circRNA	Role in Thyroid Cancer	Reference
circMAN1A2	Upregulated	[190]
circRAPGEF5	Upregulated; it acts through miR-198/FGFR1	[191]
circ_0067934	Upregulated; it improves the development of thyroid carcinoma by promoting EMT and PI3K/AKT signaling pathways	[192]
circ_0025033	Upregulated; it promotes PTC cell proliferation and invasion via sponging miR-1231 and miR-1304	[193]
Hsa_circ_0008274	Upregulated; it promotes cell proliferation and invasion involving AMPK/mTOR signaling pathway in PTC	[194]
circZFR	Regulating miR-3619-5p/CTNNB1 axis and activating Wnt/β-catenin pathway.	[195]
circRNA_102171	Upregulated; it promotes PTC progression through activating Wnt/β-catenin pathway in a CTNNBIP1-dependent way	[196]
circNUP214	Oncogenic role in PTC by acting as a sponge for miR-145, leading to upregulation of ZEB2	[197]
circ-ITCH	Upregulated; it suppresses papillary thyroid cancer progression through miR-22-3p/CBL/β-catenin pathway	[198]
hsa_circ_0004458	Promoted the progression of PTC through inhibition of miR-885-5p and activation of RAC1	[199]
chr5: 160757890-160763776-, chr12: 40696591-40697936+, chr7: 22330794-22357656-, chr21: 16386665-16415895-, chr7: 91924203-91957214+, chr2: 179514891-179516047-, chr9: 16435553-16437522-, and chr22: 36006931-36007153-	Up/downregulated	[200]
circZFR	Contributes to PTC cell proliferation and invasion by sponging miR-1261 and facilitating C8orf4 expression	[201]
hsa_circ_0137287	Downregulated in PTC tissues	[202]
hsa_circRNA_100395	Downregulated; it is related with miR-141-3p/miR-200a-3p axis in PTC tumors	[182]
hsa_circ_0058124	The hsa_circ_0058124/NOTCH3/GATAD2A axis is critical for PTC tumorigenesis and invasiveness	[203]

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
