# Peer review of "Influencers on Thyroid Cancer Onset: Molecular Genetic Basis"

_genes, 2019, doi:10.3390/genes10110913_

Round 1

Reviewer 1 Report

The study addresses an interesting and current problem about the genetic and molecular basis of thyroid cancer onset. However, I have certain objections related to the paper.

First of all, reading the paper, I have the impression that either it is some kind of patchwork. Therefore, the paper seems to be rather chaotic. I suggest you to rewrite some parts to give a coherent guideline to the readers.

Some references are not properly coherent with some affirmation in text or lack in some points, for example: “Thyroid cancer is the sixth most common cancer in women, who are three times more likely to have thyroid cancer than men, although women and men die at similar rates. About 2% of cases occur in children and teens. Overall, the 5-year survival rate of thyroid cancer is 98%. However, survival rates depend on many factors, such as the specific type of thyroid cancer and stage of disease”. Please review some references.

The authors refer to the tables and figures in the text, but these are provided only as supplementary material. Is there any reason? I think that the figures and tables should be included in the main text and supplementary material should be considered as “additional data”.

In conclusion, I recommend the major revision for the paper to be accepted

Author Response

Dear Reviewer,

Thank you for the revision of the Review “Genes-598127” entitled “Influencers on thyroid cancer onset: molecular genetic basis”.

We have written a “point-by point” letter to answer all the comments and suggestions. All the changes have been highlighted in green.

The study addresses an interesting and current problem about the genetic and molecular basis of thyroid cancer onset. However, I have certain objections related to the paper.

First of all, reading the paper, I have the impression that either it is some kind of patchwork. Therefore, the paper seems to be rather chaotic. I suggest you to rewrite some parts to give a coherent guideline to the readers.

Some references are not properly coherent with some affirmation in text or lack in some points, for example: “Thyroid cancer is the sixth most common cancer in women, who are three times more likely to have thyroid cancer than men, although women and men die at similar rates. About 2% of cases occur in children and teens. Overall, the 5-year survival rate of thyroid cancer is 98%. However, survival rates depend on many factors, such as the specific type of thyroid cancer and stage of disease”. Please review some references.

We apologize for the lack of consistency: the reason was that the list of references was wrongly linked but it has been revised, and each paragraph fixes with the proper reference. In addition, we have modified some phrases and added new ones to help the reader to understand the manuscript.

The authors refer to the tables and figures in the text, but these are provided only as supplementary material. Is there any reason? I think that the figures and tables should be included in the main text and supplementary material should be considered as “additional data”.

We put them as supplementary due to the length of some tables, but we agree that they should be included in the main text, so we thank your suggestion and currently the 5 tables and 2 figures are located after their first mention in the main text. We have changed the order to be more coherent with the text and now previous table S2 is currently Table 1 and previous table S1 is Table 2.

In conclusion, I recommend the major revision for the paper to be accepted

Reviewer 2 Report

The review by Berta Luzon-Toro is focused on the molecular genetic basis of thyroid cancer.There is no doubt that this topic is very relevant as also testified by a huge number of papers covering these issues published in the last few years.Some arguments are described in detail (mostly the genes well known by their role in thyroid carcinogenesis); on the contrary, other potential contributors to the thyroid cells transformation (ncRNAs) are just listed without critical comments

Specific comments:

FoxE1 could have an important role in thyroid cancer. This gene is not discussed at all although many papers focused on its role in cancer have been published. The molecular pathway involved in the potential progression from well differentiated carcinoma to undifferentiated and aggressive anaplastic carcinoma should be presented in more detail. The role of some specific miRNAs and lncRNAs in thyroid cancer is worthy of an extensive discussion.

Author Response

Dear Reviewer,

Thank you for the revision of the Review “Genes-598127” entitled “Influencers on thyroid cancer onset: molecular genetic basis”.

We have written a “point-by point” letter to answer all the comments and suggestions. All the changes have been highlighted in green.

The review by Berta Luzon-Toro is focused on the molecular genetic basis of thyroid cancer.There is no doubt that this topic is very relevant as also testified by a huge number of papers covering these issues published in the last few years.Some arguments are described in detail (mostly the genes well known by their role in thyroid carcinogenesis); on the contrary, other potential contributors to the thyroid cells transformation (ncRNAs) are just listed without critical comments

Specific comments:

FoxE1 could have an important role in thyroid cancer. This gene is not discussed at all although many papers focused on its role in cancer have been published.

Thank you for such suggestion. Regarding the inclusion of FoxE1 in the text, we did it on the paragraph “Other genes” and add 3 references concerning its role in thyroid cancer.

The molecular pathway involved in the potential progression from well differentiated carcinoma to undifferentiated and aggressive anaplastic carcinoma should be presented in more detail.

Apart from the different references along the text and the tables concerning the different genetic mutations associated to each type of thyroid cancer, we have included in page 10 (before the chapter of ncRNAs), a paragraph to compile the information about the molecular pathway involved in the potential progression from well differentiated carcinoma to undifferentiated and aggressive anaplastic carcinoma. Thank you for this recommendation.

The role of some specific miRNAs and lncRNAs in thyroid cancer is worthy of an extensive discussion.

Finally, we have included a deeper discussion of some miRNAs and lncRNAs, as well as a short comment about circRNAs, which are mostly exclusively linked to this type of cancer. In addition, we have updated the manuscript with the most recent articles concerning the topic of the review.

Round 2

Reviewer 1 Report

My recommendation is: accept for publication.

Reviewer 2 Report

Authors addressed my questions.

This manuscript is a resubmission of an earlier submission. The following is a list of the peer review reports and author responses from that submission.

Round 1

Reviewer 1 Report

This manuscript describes the molecular basic of thyroid cancer, especially including some noncoding RNAs that is seldom added in thyroid reviews before, which should be interesting to readers. After careful reading, I have following concerns.

1, Some parts are too short and thus failed to deliver enough information to the reader. For example, the abstract is too simple to show what’s thyroid cancer and what’s key points the author described in this manuscript.

2, The author may add the survival rate of thyroid cancer in this manuscript.

3, In the summary, the author may add some future prospects.

4, It will be more friendly to readers if the author could provide a figure showing the progress of thyroid and all involved genetic alterations.

Author Response

Dear Reviewer,

Thank you for the review of the Review “genes-548634” entitled “Influencers on thyroid cancer onset: molecular genetic basis”.

We have written a “point-by point” letter to answer all comments and suggestions from you. All changes have been marked in the manuscript using Track Changes system. New References have been marked in yellow. In addition, a new figure (S1) and a new table (S1) have been included on this version.

Comments and Suggestions for Authors

Reviewer 1

This manuscript describes the molecular basic of thyroid cancer, especially including some noncoding RNAs that is seldom added in thyroid reviews before, which should be interesting to readers. After careful reading, I have following concerns.

1, Some parts are too short and thus failed to deliver enough information to the reader. For example, the abstract is too simple to show what’s thyroid cancer and what’s key points the author described in this manuscript.

Thanks for this suggestion. We have extended the abstract explaining in more detail what’s thyroid cancer and what key points we afford on the manuscript (lines 11-18).

2, The author may add the survival rate of thyroid cancer in this manuscript.

We have added this relevant information in lines 23-27.

3, In the summary, the author may add some future prospects.

Thanks for this observation. We have included different future prospects (lines 235-257) with their references (highlighted in yellow).

4, It will be more friendly to readers if the author could provide a figure showing the progress of thyroid and all involved genetic alterations.

We have included a new figure (Figure S1), mentioned in line 62. Then, previous Figure S1 is now named Figure S2.

Reviewer 2 Report

Influencers on Thyroid Cancer Onset: Molecular Genetic Basis

The authors present the molecular genetic basis involved in the different forms of thyroid cancer. Initially, the authors described the different histological subtypes of this cancer and, successively, described the master genetic alterations associated with the different form of thyroid cancer. 

Overall the Review is interesting and clearly written. However there are some minor comments that needed to be addressed:

·     In the Introduction the authors argue that Non-medullary thyroid cancer (NMTC) is divided into fours groups. The authors should highlight the differences betweenanaplastic (3) and undifferentiated (4) thyroid cancer. In addition, an appropriate reference must be added.

·     Authors could report in the main manuscript a novel Table with the most relevant genes (RET, BRAF, STAT3, P53…), their alterations and the relative frequencies in thyroid cancer. 

·     In the Tables S1 and S2 all the References must be added.

·     Integrated Genomic Characterization of Papillary Thyroid Carcinoma (Cell, 2014) should be referenced and reported in the “Molecular Genetics of Thyroid Cancer” paragraph and in the Discussion.In particular, needs more depth of discussion about the potential translatable impact of these findings.

·     Moreover, the authors should also provide a comment on the other mutation driver of thyroid cancer, such as mutation in TERT promoter, ALK, NTRK3.

·     The Discussion is not speculative but is rather repetitive.

Author Response

Dear Reviewer,

Thank you for the review of the Review “genes-548634” entitled “Influencers on thyroid cancer onset: molecular genetic basis”.

We have written a “point-by point” letter to answer all comments and suggestions from you. All changes have been marked in the manuscript using Track Changes system. New References have been marked in yellow. In addition, a new figure (S1) and a new table (S1) have been included on this version.

Comments and Suggestions for Authors

Reviewer 2

The authors present the molecular genetic basis involved in the different forms of thyroid cancer. Initially, the authors described the different histological subtypes of this cancer and, successively, described the master genetic alterations associated with the different form of thyroid cancer. 

 Overall the Review is interesting and clearly written. However, there are some minor comments that needed to be addressed:

 ·     In the Introduction the authors argue that Non-medullary thyroid cancer (NMTC) is divided into fours groups. The authors should highlight the differences between anaplastic (3) and undifferentiated (4) thyroid cancer. In addition, an appropriate reference must be added.

 We apologize for this mistake. We reorganize the classification, with “poorly differentiated” instead of “undifferentiated” (lines 32-34) and we added a proper reference (highlighted in yellow).

·     Authors could report in the main manuscript a novel Table with the most relevant genes (RET, BRAF, STAT3, P53…), their alterations and the relative frequencies in thyroid cancer. 

Regarding the suggested table with the most relevant genes, their alterations and frequencies in thyroid cancer, we must consider that:

1) There are many different types of thyroid cancer (sporadic, familiar, syndromic...).

2) Each of those types, present a huge number of associated variants published to date (some have been confirmed as pathogenic and some other not or not yet).

Then, we consider that it was going to be easier for the reader to have:

-- a table indicating the most relevant genes, indicating their location on the genome, their main type of alterations and their classification as somatic or germline mutations, all properly referenced.

--a footnote with the main databases available to access to all RET variants “with clinical relevance to date”, which is the most relevant gene implicated in the disease, indicating their number registered on each database until the last access (22/07/2019). 

·     In the Tables S1 and S2 all the References must be added.

All references for Table S2 (previous Table S1) were included and highlighted in yellow.

Regarding the references for Table S3 (previous Table S2), references 44 to 49 were included in a combined cell by meaning that all these references compile all the miRNAs included in the table.  

·     Integrated Genomic Characterization of Papillary Thyroid Carcinoma (Cell, 2014) should be referenced and reported in the “Molecular Genetics of Thyroid Cancer” paragraph and in the Discussion. In particular, needs more depth of discussion about the potential translatable impact of these findings.

 Thanks for such interesting recommendation. We have included some comments about this paper on the “Molecular Genetics of Thyroid Cancer” paragraph (lines 72-80) and in the “Conclusions” section (lines 276-284).

·     Moreover, the authors should also provide a comment on the other mutation driver of thyroid cancer, such as mutation in TERT promoter, ALK, NTRK3.

We have extended the information that was already included for TERT promoter (see section 5 “RAS”, lines 169-174). Regarding ALK and NTRK3, we have included them in the “Other genes” section (lines 193-200).

·     The Discussion is not speculative but is rather repetitive.

The “Conclusions” section, has been currently changed to “Conclusions and Future Perspectives”, in order to avoid being repetitive and thus, to discuss some new approaches and findings related with the current and future perspectives in the management of thyroid cancer patients.

Round 2

Reviewer 1 Report

All my concerns are addressed. I think it fits to publish now.